# The common *Sting1 HAQ, AQ* alleles rescue CD4 T cellpenia, restore T-regs, and prevent *SAVI (N153S)* inflammatory disease in mice

Alexandra a Aybar-Torres[1], Lennon A Saldarriaga[1], Ann T Pham[1], Amir M Emtiazjoo[1], Ashish K Sharma[2], Andrew j Bryant[1], Lei Jin[1]*

[1]Division of Pulmonary, Critical Care and Sleep Medicine, Department of Medicine, University of Florida, Gainesville, United States; [2]Division of Vascular Surgery & Endovascular Therapy, Department of Surgery, University of Florida, Gainesville, United States

*For correspondence:
lei.jin@medicine.ufl.edu

**Abstract** The significance of *STING1* gene in tissue inflammation and cancer immunotherapy has been increasingly recognized. Intriguingly, common human *STING1* alleles R71H-G230A-R293Q (*HAQ*) and G230A-R293Q (*AQ*) are carried by ~60% of East Asians and ~40% of Africans, respectively. Here, we examine the modulatory effects of *HAQ, AQ* alleles on STING-associated vasculopathy with onset in infancy (SAVI), an autosomal dominant, fatal inflammatory disease caused by gain-of-function human *STING1* mutations. CD4 T cellpenia is evident in SAVI patients and mouse models. Using *Sting1* knock-in mice expressing common human *STING1* alleles *HAQ, AQ,* and *Q293*, we found that *HAQ, AQ,* and *Q293* splenocytes resist STING1-mediated cell death ex vivo, establishing a critical role of STING1 residue 293 in cell death. The *HAQ/SAVI(N153S)* and *AQ/SAVI(N153S)* mice did not have CD4 T cellpenia. The *HAQ/SAVI(N153S), AQ/SAVI(N153S)* mice have more (~10-fold, ~20-fold, respectively) T-regs than *WT/SAVI(N153S)* mice. Remarkably, while they have comparable TBK1, IRF3, and NFκB activation as the *WT/SAVI*, the *AQ/SAVI* mice have no tissue inflammation, regular body weight, and normal lifespan. We propose that STING1 activation promotes tissue inflammation by depleting T-regs cells in vivo. Billions of modern humans have the dominant *HAQ, AQ* alleles. STING1 research and STING1-targeting immunotherapy should consider *STING1* heterogeneity in humans.

## eLife assessment

This study describes **useful** mouse models of knock-ins of human STING1 variants and an assessment of these variants' action in mouse immune cells. While the implications of the variants in the inflammatory response are of significant interest, limitations are still found in the authors' interpretation and conclusions made, and the evidence for the conclusion remains **incomplete**.

## Introduction

STING1 drives cytosolic DNA-induced type I IFNs production (*Barber, 2015*). Recent research revealed that STING1 promotes inflammation in a variety of inflammatory diseases, including nonalcoholic fatty liver disease, nonalcoholic steatohepatitis, kidney injury, neurodegenerative diseases, cardiovascular diseases, obesity, diabetes, and aging (*Skopelja-Gardner et al., 2022; Bai and Liu, 2021; Li et al., 2023; Gulen et al., 2023; Szego et al., 2022; Gao et al., 2023b; Yan et al., 2022;*

*Gao et al., 2023a*). The type I IFNs-independent function of STING1 has also emerged (*Gao et al., 2023a*; *Yamashiro et al., 2020*; *Wu et al., 2020*; *Pham et al., 2024*). For example, initially described as a type I interferonopathy (*Liu et al., 2014*), recent studies in STING-associated vasculopathy with onset in infancy (SAVI) mouse models showed that SAVI is largely independent of type I IFNs (*Luksch et al., 2019*; *Stinson et al., 2022*; *Warner et al., 2017*; *Frémond et al., 2021*). In a *Sting1 N153S* mouse model of SAVI, crossing *N153S* mice to IRF3/IRF7, and IFNAR1 knockout mice, N153S mice still developed spontaneous lung diseases (*Luksch et al., 2019*). JAK inhibitors were used to block type I IFNs signaling for SAVI patients with mixed success (*Frémond et al., 2021*; *Kim et al., 2022*; *Volpi et al., 2019*; *Tang et al., 2020*). For example, in a review of JAK inhibition in 18 SAVI patients, incomplete response to treatment happened in 7/18 (38%) of patients (*Dai et al., 2020*). Furthermore, two patients died of respiratory failure despite this treatment (*Dai et al., 2020*). Both radioresistant parenchymal and/or stromal cells and hematopoietic cells influence SAVI pathology in mice (*Gao et al., 2024*; *Gao et al., 2022*). The observation is important because it predicts that allogeneic stem cell transplantation may not work in human SAVI patients. Indeed, lung transplantations did not show improvement in SAVI patients (*Kim et al., 2022*; *Picard et al., 2016*). Patients died at 3- and 9 months post-lung transplant (*Kim et al., 2022*; *Picard et al., 2016*). How STING1 drives inflammation in vivo, independent of type I IFNs, remains unknown. Consequently, SAVI has no curative care.

Characterized as an innate immune sensor, STING1 expression is, paradoxically, high in CD4 T cells (*Liu et al., 2014*; *Jin et al., 2013*). Furthermore, STING1 activation kills mouse and human CD4 T cells ex vivo (*Larkin et al., 2017*; *Cerboni et al., 2017*; *Kuhl et al., 2023*). SAVI patients and mouse models had CD4 T cellpenia (*Liu et al., 2014*; *Kuhl et al., 2023*). STING1 was first discovered as MPYS for its cell growth inhibition and cell death function in mouse B lymphoma cells (*Jin et al., 2008*). STING1-mediated cell death is cell type dependent. For example, while STING1 activation kills human endothelial cells, primary and cancerous T cells, it does not kill mouse MEFs, BMDCs, or BMDMs (*Gulen et al., 2017*; *Kabelitz et al., 2022*; *Murthy et al., 2020*). Second, STING1-mediated cell death is type I IFNs-independent (*Kuhl et al., 2023*; *Gulen et al., 2017*; *Murthy et al., 2020*). Multiple cell death pathways, that is apoptosis, necroptosis, pyroptosis, ferroptosis, and PANoptosis, are proposed (*Murthy et al., 2020*; *Li et al., 2021*; *Song et al., 2022*). Last, the in vivo biological significance of STING1-mediated CD4 T cell death is not clear (*Kuhl et al., 2023*; *Murthy et al., 2020*). In humans, SAVI patients with constitutively activated STING1 have low CD4 T cell numbers (*Liu et al., 2014*), and type I IFNs are dispensable for STING1-mediated human CD4 T cell death (*Kuhl et al., 2023*). Different from SAVI mice, SAVI patients (*N154S* or *V155M*) had normal counts of CD8 T and B cells (*Liu et al., 2014*).

The human *STING1* gene is highly heterogeneous (*Jin et al., 2011*; *Patel et al., 2017a*). Approximately 50% of people in the U.S. carry at least one copy of non-*WT STING1* allele (*Jin et al., 2011*). Among them, the R71H-G230A-R293Q (*HAQ*) is the second most common *STING1* allele carried by ~23% of people in the U.S. (*Jin et al., 2011*). However, in East Asians, *WT/HAQ* (34.3%), not *WT/WT* (22.0%), is the most common *STING1* genotype (*Patel et al., 2017a*). Critically, the *HAQ* allele was positively selected in modern humans outside Africa *Mansouri et al., 2022*. Anatomically modern humans outside Africa are descendants of a single Out-of-Africa Migration 50,000~70,000 years ago. ~1.4% of Africans have the *HAQ* allele, while ~63.9% of East Asians are *HAQ* carriers (*Mansouri et al., 2022*). Haplotype analysis revealed that *HAQ* was derived from G230A-R293Q (*AQ*) allele (*Mansouri et al., 2022*). Importantly, the *AQ* allele was negatively selected outside Africa (*Mansouri et al., 2022*). Approximately 40.1% of Africans are *AQ* carriers, while ~0.4% of East Asians have the *AQ* allele (*Mansouri et al., 2022*). *STING1* alleles often have a dominant negative effect likely because the protein STING1 exists as a homodimer (*Jin et al., 2008*; *Patel and Jin, 2019*). SAVI is an autosomal dominant inflammatory disease (*Liu et al., 2014*). *WT/HAQ* individuals had reduced Pneumovax23-induced antibody responses compared to *WT/WT* individuals (NCT02471014) (*Sebastian et al., 2020*). Notably, *AQ* responds to CDNs and produces type I IFNs in vivo and in vitro (*Mansouri et al., 2022*; *Yi et al., 2013*; *Patel et al., 2017b*), but the *AQ* allele was negatively selected in non-Africans (*Mansouri et al., 2022*). In contrast, the *HAQ* allele, defective in CDNs-type I IFNs responses (*Jin et al., 2011*; *Patel et al., 2017a*; *Sebastian et al., 2020*; *Nissen et al., 2018*; *Ruiz-Moreno et al., 2018*), was positively selected in non-Africans (*Mansouri et al., 2022*), indicating that the CDNs-type I IFNs independent function of STING1 was essential for the survival of early modern humans outside of Africa.

In this study, we discovered, surprisingly, that the *HAQ, AQ* splenocytes are resistant to STING1-mediated cell death. We generated *HAQ/SAVI(N153S)* and *AQ/SAVI(N153S)* mice and found that the *HAQ, AQ* alleles prevent CD4 T cellpenia, increasing/restoring T-regs and alleviating/stopping tissue inflammation in SAVI mice, thus providing evidence for the in vivo significance of type I IFNs-independent, STING1-mediated cell death and potential *AQ*-based curative therapy for SAVI patients.

## Results

### STING1 activation kills mouse spleen CD4, CD8 T, and CD19 B cells ex vivo

We first used the synthetic non-CDNs STING1 agonist diABZI (*Ramanjulu et al., 2018*) to induce lymphocyte death because diABZI induces cell death without the need for lipid transfection or detergent for cell permeabilization (*Kabelitz et al., 2022*; *Messaoud-Nacer et al., 2022*) and diABZI is in clinical trials (NCT05514717). Splenocytes from C57BL/6 N mice were treated with diABZI in culture, and cell death was determined by Annexin V and Propidium Iodide stain. Splenocyte cell death could be detected at 5 hr post diABZI treatment (*Figure 1—figure supplement 1A*). Dosage responses showed that ~25 ng/ml diABZI could kill 70% of splenocytes (*Figure 1—figure supplement 1B*). Similarly, STING1 agonists DMXAA and synthetic CDNs RpRpss-Cyclic di-AMP killed mouse spleen CD4, CD8 T cells, and CD19 B cells (*Figure 1A*). Thus, STING1 activation readily induces mouse lymphocyte death ex vivo.

### TBK1 activation is required for STING1-mediated mouse spleen cell death ex vivo

STING1 activation can lead to apoptosis, pyroptosis, necroptosis, or ferroptosis (*Kuhl et al., 2023*; *Jin et al., 2008*; *Gulen et al., 2017*; *Kabelitz et al., 2022*; *Murthy et al., 2020*; *Li et al., 2021*; *Song et al., 2022*; *Messaoud-Nacer et al., 2022*; *Tang et al., 2016*). We then treated mouse splenocytes with apoptosis, pyroptosis, necroptosis inhibitors, STING1 inhibitors H-151, C-176, and palmitoylation inhibitor 2-bromopalmitate (2 BP), followed by diABZI stimulation. Inhibitors for NLRP3 (MCC950), RIPK1 (Necrostatin-1), RIPK3 (GSK872), Caspase-1 (VX-795), Caspase-3 (Z-DEVD-FMK), Caspase 1,3,8,9 (Q-VD-Oph), ferroptosis (liproxstatin-1) did not affect diABZI-induced splenocyte cell death ex vivo (*Figure 1—figure supplement 1B, C*). The STING1 inhibitors H-151, C-176, and 2 BP also could not prevent diABZI-induced cell death (*Figure 1—figure supplement 1C*), although they inhibited diABZI-induced IFNβ production (*Figure 1—figure supplement 1D*). Instead, the TBK1 inhibitor BX-795 abolished diABZI-induced splenocyte death (*Figure 1—figure supplement 1C*).

BX-795 is a multi-kinase inhibitor, including 3-phosphoinositide-dependent protein kinase 1 (PDK1) and TBK1 ($IC_{50s}$ = 6 and 11 nM, respectively). However, the treatment of PDK1 inhibitor GSK2334470 (IC50=10 nM) did not prevent diABZI-induced splenocyte death (*Figure 1B*). In contrast, GSK8612, a highly potent and selective inhibitor for TBK1, prevented diABZI-induced splenocyte death (*Figure 1B*). Thus, TBK1 activation is likely critical for STING1-mediated splenocyte cell death ex vivo.

### *HAQ, AQ, Q293 STING1* knock-in mouse splenocytes are resistant to STING1-mediated cell death ex vivo

*HAQ* and *AQ* are common human *STING1* alleles (*Jin et al., 2011*; *Patel et al., 2017a*; *Mansouri et al., 2022*). Previously, we reported that *HAQ* knock-in mice are defective in CDNs-induced immune responses, while CDNs responses in *AQ* knock-in mice are similar to WT mice (*Mansouri et al., 2022*). We treated splenocytes from *HAQ* and *AQ* mice with diABZI ex vivo and found, surprisingly, that both *HAQ* and *AQ* splenocytes were resistant to diABZI-induced cell death (*Figure 1C and D*). In comparison, *IFNAR1*[-/-] splenocytes were killed by diABZI, confirming that STING1-mediated lymphocytes death are type I IFNs-independent (*Figure 1C and D*; *Kuhl et al., 2023*; *Gulen et al., 2017*; *Murthy et al., 2020*).

*HAQ* and *AQ* share the common A230 and Q293 residues changes. We thus generated a *Q293 Sting1* knock-in mouse. Notably, the Q293 splenocytes were resistant to STING1 agonists 2'3'-cGAMP, RpRpss-Cyclic di-AMP, and diABZI-induced cell death (*Figure 1E and F*). Thus, the residue 293 of STING1 is critical for its cell death function.

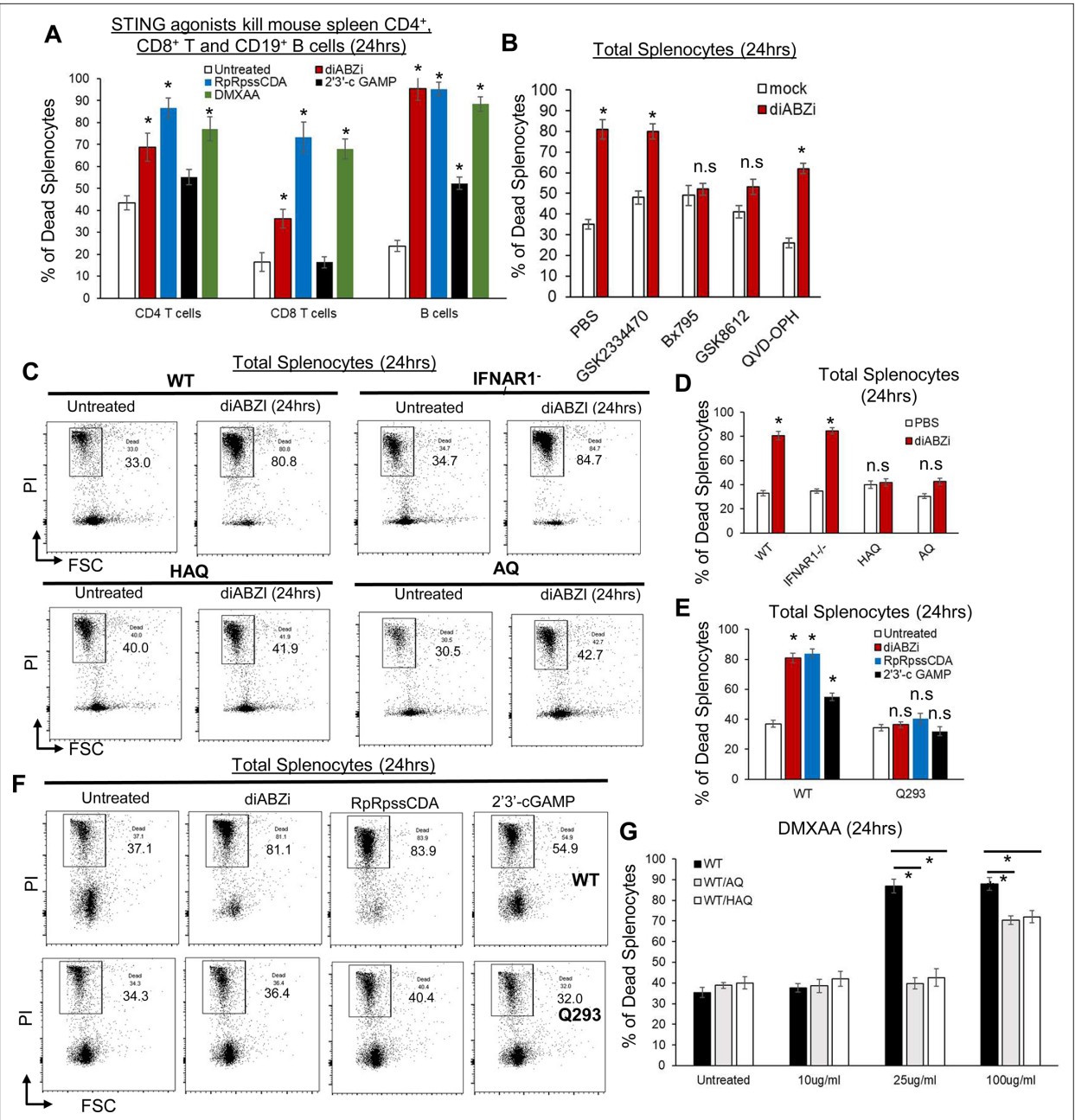

**Figure 1.** Splenocytes from *HAQ, AQ,* and *Q293* mice are resistant to STING1-mediated cell death ex vivo. (**A**) C57BL/6 N splenocytes were treated directly (no transfection) with diABZI (100 ng/ml), RpRpss-Cyclic di-AMP (5 µg/ml) or 2'3'-cGAMP (10 µg/ml), DMXAA (25 µg/ml) for 24 hr in culture. CD4, CD8 T cells and CD19 B cells death were determined by PI staining. (**B**). Splenocytes from C57BL/6 N mice were pre-treated with indicated small molecules, GSK2334470 (1.25 µM), GSK8612 (2.5 µM), Bx-795 (0.5 µM), QVD-OPH (25 µM) for 2 hrs. diABZI (100 ng/ml) was added in culture for another 24 hr. Dead cells were determined by PI staining. (**C–D**). Flowcytometry of *HAQ, AQ, IFNAR1*[-/-] or C57BL/6 N splenocytes treated with diABZI (100 ng/ml) for 24 hrs. Cell death was determined by PI staining. (**E–F**). *Q293* or the *WT* littermates splenocytes were treated with diABZI (100 ng/ml), RpRpss-Cyclic di-AMP (5 µg/ml) or 2'3'-cGAMP 10 µg/ml for 24 hr. Cell death was determined by PI staining. (**G**). *WT/HAQ, WT/AQ,* or *WT/WT* littermates splenocytes were treated with DMXAA (10, 25 or 100 µg/ml) for 24 hr. Cell death was determined by PI staining. Data are representative of three independent experiments. Graphs represent the mean with error bars indication s.e.m. *p* values are determined by one-way ANOVA Tukey's multiple comparison test (**A, E, G**) or unpaired student T-test (**B, D**) * p<0.05. n.s: not significant.

The online version of this article includes the following figure supplement(s) for figure 1:

**Figure supplement 1.** Bx-795 inhibits diABZI-induced mouse splenocyte death.

### WT/HAQ, WT/AQ mouse splenocytes are partially resistant to STING1-mediated cell death ex vivo

WT/HAQ (34.3%) is the most common human STING1 genotype in East Asians, while WT/AQ (28.2%) is the 2nd most common STING1 genotype in Africans (*Patel et al., 2017a*). We generated WT/HAQ, WT/AQ mice and treated their splenocytes with mouse STING1 agonist DMXAA. WT/HAQ and WT/AQ splenocytes were protected from 25 μg/ml DMXAA-induced cell death (*Figure 1G*). A total of 100 μg/ml DMXAA could kill WT/HAQ and WT/AQ splenocytes, albeit less than WT/WT cells (*Figure 1G*). Thus, the HAQ and AQ alleles are dominant and likely impact STING1 activation even in heterozygosity.

### STING1 activation kills primary human CD4 T cells but not CD8 T or CD19 B cells

STING1 agonists-based clinical trials in humans have been disappointing (NCT02675439, NCT03010176, NCT05514717; *Meric-Bernstam et al., 2023*; *Meric-Bernstam et al., 2022*). We showed that the human STING1 gene might undergo natural selection during the out-of-Africa migration (*Mansouri et al., 2022*) sensitive to evolutionary pressure. Thus, we investigated STING1-mediated death in primary human lymphocytes.

Human explant lung cells from the WT(R232)/WT(R232) donors were treated with STING1 agonists 2'3 c GAMP, RpRpss-Cyclic di-AMP, diABZI for 24 hr in culture. Lymphocyte cell death was determined by Propidium Iodide staining. Different from mouse lymphocytes, diABZI and RpRpss-Cyclic di-AMP killed human CD4 T but not CD8 T or CD19 B cells (*Figure 2A and B*). Human CD8 T and CD19 B cells are resistant to 500 ng/ml diABZI-induced cell death (*Figure 2—figure supplement 1A*).

### WT/HAQ human CD4 T cells are resistant to low doses of diABZI-induced cell death

WT/HAQ mouse splenocytes are resistant to low-dose diABZI-induced cell death (*Figure 1G*). To extend our observation into primary human T cells, we obtained lung explants from WT/WT and WT/HAQ individuals (*Figure 2—figure supplement 1B*) and treated them with diABZI in culture. 25 ng/ml diABZI killed WT/WT, but not WT/HAQ, human lung CD4 T cells (*Figure 2C*).

### diABZI induces cell death in STING1-KO human THP-1 cells reconstituted with WT human STING1 (R232) but not HAQ, AQ or Q293 human STING1 allele

To further determine cell death influenced by human STING1 alleles HAQ, AQ, and Q293, we used the STING1-KO THP-1 cell line because STING1 agonist induces type I IFNs and cell death in STING1-KO THP-1 cells expressing WT human STING1 (*Song et al., 2022*; *Figure 2—figure supplement 1C and D*). We, thus, generated stable THP-1 STING1-KO lines expressing HAQ, AQ, WT, or Q293 STING1 allele. Cell death was determined by Annexin-V staining. diABZI killed THP-1 STING1-KO lines expressing WT but not HAQ, AQ, or Q293 STING1 allele (*Figure 2D and E*). No cell death was induced in the Q293 THP-1 cells stimulated by 20–200 ng/ml of diABZI (*Figure 2F*). diABZI also did not induce STING1 activation in Q293 THP-1 cells (*Figure 2G*). Notably, 50 ng/ml diABZI induced p-IRF3 activation and type I IFNs in AQ THP-1 cells but not HAQ THP-1 cells (*Figure 2H and I*), indicating that the STING1-cell death and STING1-IRF3-Type I IFNs pathways can be uncoupled.

### HAQ and AQ alleles rescue the lymphopenia and suppress myeloid cell expansion in SAVI(N153S) mice

The in vivo significance of the STING1/MPYS-cell death is unclear. Furthermore, multiple cell death pathways, that is apoptosis, necroptosis, pyroptosis, ferroptosis, and PANoptosis, are proposed (*Murthy et al., 2020*; *Li et al., 2021*; *Song et al., 2022*). The uncertainty likely results from studies using different cell types (primary cells vs cancer cell lines); species (human vs mouse); STING1 agonists (cGAMP, which requires cell permeabilization by detergents or lipid transfection, vs diABZi, DMXAA that can directly cross the membrane; *Larkin et al., 2017*; *Cerboni et al., 2017*; *Gulen et al., 2017*; *Kabelitz et al., 2022*; *Wu et al., 2019*). Critically, which mechanism is relevant in vivo, causing T

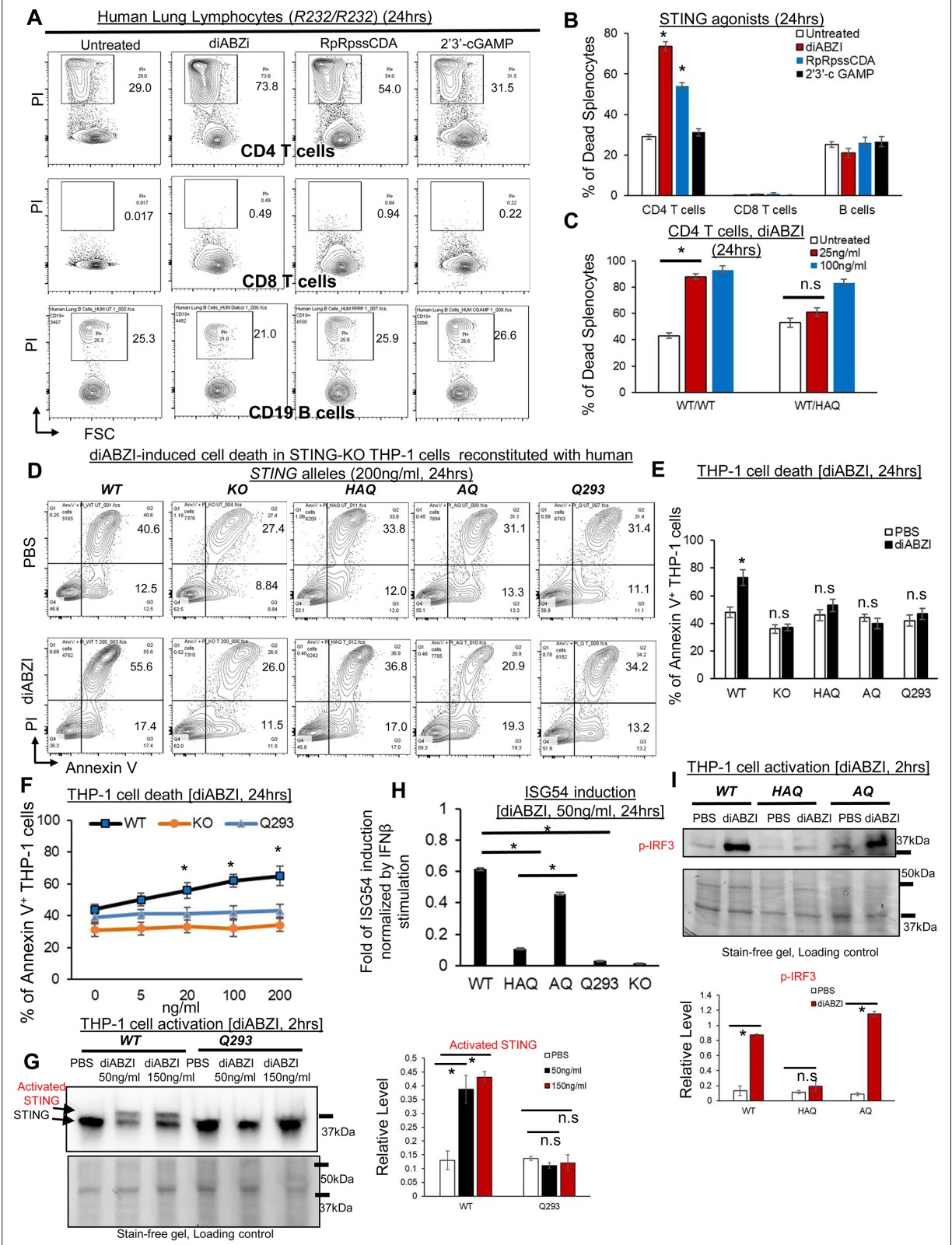

**Figure 2.** *HAQ, AQ, and Q293 human cells are resistant to STING1 agonists-induced death.* (**A–B**) Total human Lung cells from *WT/WT* individuals were treated with diABZi (100 ng/ml) for 24 hr. Cell death in CD4, CD8 T cells and CD19 B cells were determined by PI staining. (**C**). Total lung cells from a *WT/HAQ* (2 individuals) and a *WT/WT* (3 individuals) were treated with diABZi (25, 100 ng/ml) for 24 hr. Cell death in CD4 T cells was determined by PI staining. (**D–E**). STING1-KO THP-1 cells (Invivogen,, cat no. thpd-kostg) were stably reconstituted with human *WT (R232), HAQ, AQ, Q293.*

*Figure 2 continued on next page*

*Figure 2 continued*

Cells were treated with diABZI (200 ng/ml) in culture for 24 hr. Dead cells were determined by Annexin V staining. (**F**). STING1-KO THP-1 cells stably reconstituted with human *WT (R232), Q293* were treated with indicated dose of diABZI for 24 hr in culture. Dead cells were determined by Annexin V staining. (**G**). STING1-KO THP-1 cells stably reconstituted with human *WT (R232), Q293* were treated with indicated dose of diABZI for 2hs in culture. STING1 activation was detected by anti-STING1 antibody (Proteintech, #19851–1-AP). (**H**). STING1-KO THP-1 cells stably reconstituted with human *WT (R232), HAQ, AQ, Q293* were treated with 50 ng/ml diABZI in culture for 24 hr. ISG-54 reporter luciferase activity was determined in cell supernatant and normalized to 10 ng/ml IFNβ-stimulated ISG-54 luciferase activity. (**I**). STING1-KO THP-1 cells stably reconstituted with human *WT (R232), HAQ, AQ* were treated with 50 ng/ml diABZI in culture for 2 hr. STING1 and IRF3 activation were determined by anti-STING1 antibody and anti-p IRF3 antibody (CST, Ser396, clone 4D4G). Densitometry was determined by ImageLab 5. Data are representative of three independent experiments. Graphs represent the mean with error bars indication s.e.m. p values determined by one-way ANOVA Tukey's multiple comparison test (**B, C, F, H, G**) or unpaired student T-test (**E, I**). * p<0.05, n.s: not significant.

The online version of this article includes the following figure supplement(s) for figure 2:

**Figure supplement 1.** STING1 activation in primary human cells and THP-1 cells reconstituted with *WT* human *STING1*.

cellpenia is not known. To clarify the in vivo significance and mechanisms of STING1-mediated cell death, we turned to SAVI mice.

SAVI is an autosomal dominant, inflammatory disease caused by one copy of a gain-of-function *STING1* mutant (*WT/SAVI*) *Liu et al., 2014*). CD4 T cellpenia was found in SAVI patients and SAVI mouse models (*Liu et al., 2014*; *Luksch et al., 2019*. STING1 activation in SAVI mice is independent of ligands and happens in vivo. We thus generated *HAQ/SAVI(N153S)* and *AQ/SAVI(N153S)* mice aiming to establish the in vivo significance and mechanism of STING1-cell death.

First, *HAQ/SAVI(N153S)* and *AQ/SAVI(N153S)* mice had reduced splenomegaly compared to *WT/SAVI(N153S)* mice though their spleens were still larger than the littermates *WT/HAQ* and *WT/AQ* (*Figure 3A and B*). Next, *HAQ/SAVI(N153S)* and *AQ/SAVI(N153S)* mice had similar spleen B cells and CD4 T cell numbers as the *WT/HAQ, WT/AQ* littermates (*Figure 2B and C*). Their CD8$^+$ T cells were lower than their *WT* littermates but much higher than the *WT/SAVI(N153S)* mice (*Figure 3D*). Third, spleen myeloid cell numbers, that is neutrophils, Ly6C$^{hi}$ monocytes and F4/80 macrophages, were all reduced by half compared to *WT/SAVI(N153S)* mice (*Figure 3E, H*). Last, splenocytes from *HAQ/SAVI, AQ/SAVI* mice were partially resistant to diABZI, DMXAA-induced cell death ex vivo (*Figure 3—figure supplement 1*). Notably, the *HAQ/SAVI(N153S)* and *AQ/SAVI(N153S)* mice also had restored bone marrow monocytes (*Figure 4—figure supplement 1*). Thus, *HAQ* and *AQ* alleles prevent lympho-penia and suppress myeloid cell expansion in *SAVI(N153S)* mice.

## The *HAQ* allele alleviates and the *AQ* allele prevents *SAVI(N153S)* disease in mice

*SAVI(N153S)* disease is characterized by early onset, failure to thrive (low body weight), persistent lung inflammation, decreased lung function, and young death in humans and mouse models *Liu et al., 2014*; *Frémond et al., 2021*; *Wu et al., 2019*; *Motwani et al., 2019*. The *HAQ/SAVI(N153S)* mice weighed more and had an improved lifespan than the *WT/SAVI(N153S)* mice (*Figure 4A and B*). The lifespan, airway resistance, and tissue inflammation (lung, liver) were also improved in *HAQ/SAVI(N153S)* mice compared to the *WT/SAVI(N153S)* mice (*Figure 4C, D, J and K*). However, the pulmonary artery pressure was still elevated in *HAQ/SAVI(N153S)* mice (*Figure 4E*). Remarkably, the *AQ/SAVI(N153S)* mice had similar body weight and lifespan as the *WT/AQ* mice (*Figure 4F and G*). The airway resistance, pulmonary artery pressure, and tissue inflammation in *AQ/SAVI(N153S)* were similar to the *WT/AQ* littermates (*Figure 4H, I, J and K*). Thus, the *HAQ* allele alleviates and the *AQ* allele prevents inflammatory SAVI disease in mice. Interestingly, lungs from *HAQ/SAVI, AQ/SAVI* had similarly reduced infiltration of Ly6G$^+$CD11B$^+$ neutrophils and Ly6G$^-$Ly6C$^+$CD11B$^+$ inflammatory mono-cytes compared to *WT/SAVI* mice (*Figure 4—figure supplement 1*).

## diABZI induces similar STING1, TBK1, IRF3, NFκB activation in the *AQ/SAVI(N153S)* and *WT/SAVI(N153S)* bone-marrow-derived-macrophage (BMDM)

SAVI was characterized as type I interferonopathy (*Frémond et al., 2021*). However, several studies showed that type I IFN signaling and IRF3 activation were dispensable for SAVI disease (*Luksch et al., 2019*; *Stinson et al., 2022*; *Gao et al., 2022*; *Motwani et al., 2019*). *AQ* allele prevents SAVI disease

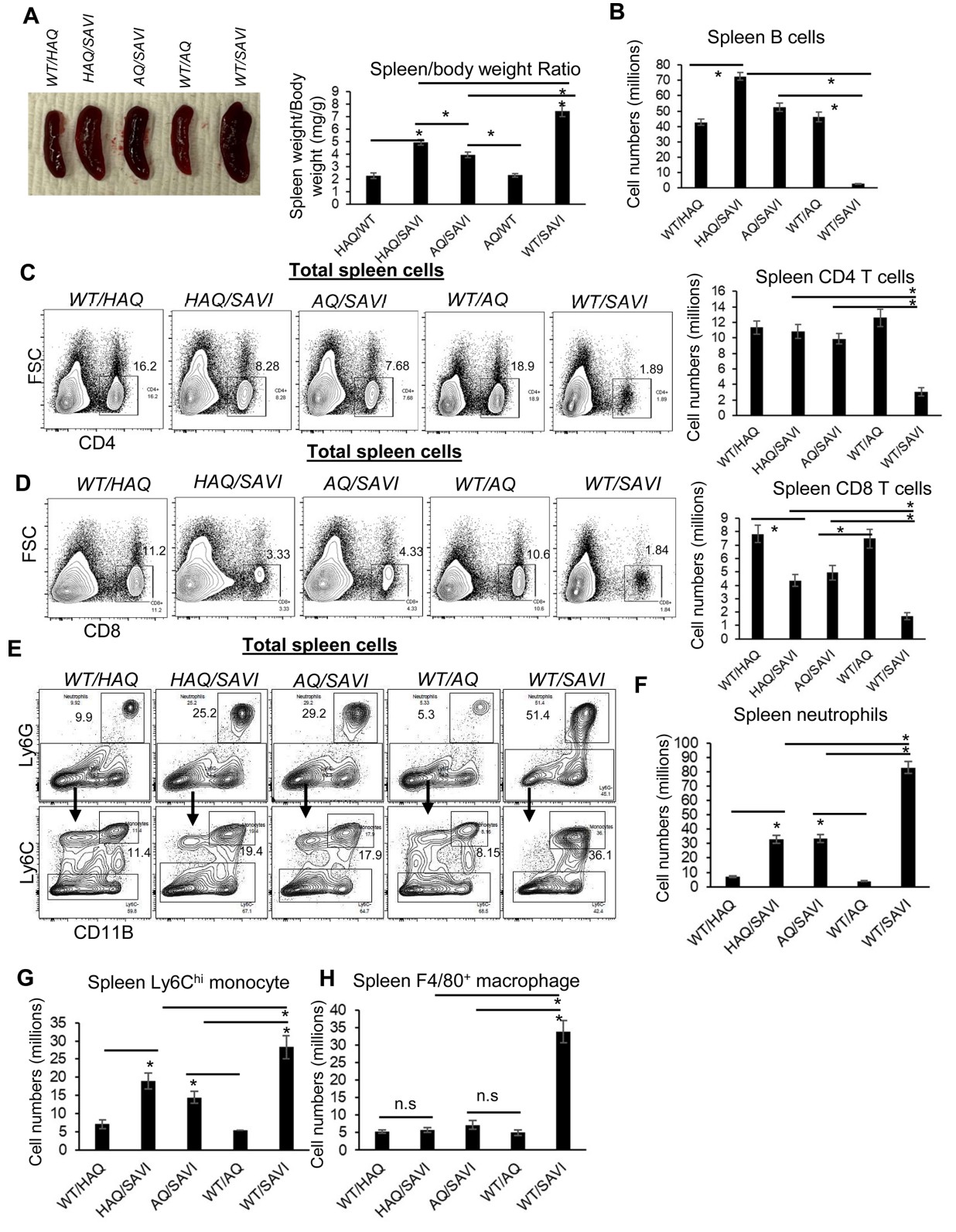

**Figure 3.** *HAQ* and *AQ* rescue the lymphopenia and suppress myeloid cell expansion in *SAVI (N153S)* mice. (**A**) The size and weight of spleens from *WT/HAQ, HAQ/SAVI, AQ/SAVI, WT/AQ, WT/SAVI*. (**B–D**). Spleen CD19+ B cells, CD4, CD8 T cells were determined in the indicated mice by Flow. (**E–H**). Spleen Ly6G+ neutrophils, Ly6Chi monocytes and F4/80+ macrophage was determined in the indicated mice by Flow. Data are representative of three

*Figure 3 continued on next page*

Figure 3 continued

independent experiments. n=3–5 mice/group. Graphs represent the mean with error bars indication s.e.m. p values are determined by one-way ANOVA Tukey's multiple comparison test. * p<0.05, n.s: not significant.

The online version of this article includes the following figure supplement(s) for figure 3:

**Figure supplement 1.** *Splenocytes from HAQ/SAVI, AQ/SAVI partially resist to STING1-activation-induced cell death* ex vivo.

**Figure supplement 2.** *HAQ and AQ restore bone marrow monocytes in SAVI (N153S) mice.*

(*Figure 4*). However, diABZI-treated *AQ/SAVI(N153S)* and *WT/SAVI(N153S)* BMDM had similar TBK1-IRF3 activation and IFNβ production (*Figure 5A and B*). diABZI treatment caused IκBα degradation, and similar TNF production in *WT/SAVI* and *AQ/SAVI* BMDM (*Figure 5C and D*). Furthermore, diABZI activation led to STING1 protein degradation in *WT/SAVI* and *AQ/SAVI* BMDM (*Figure 5E*). Last, using cleavable crosslinker dithiobis succinimidyl propionate (DSP), we showed that STING1 in *WT/SAVI, AQ/SAVI* BMDM forms a similar dimer in situ (*Figure 5F*). Thus, the *AQ/SAVI* BMDM had similar STING1 degradation, TBK1, IRF3, NFκB activation, and dimerization as the *WT/SAVI* BMDM.

### *The HAQ allele increased, and the AQ allele restored T-regs in SAVI(N153S) mice*

IFNγ was proposed to drive SAVI disease (*Stinson et al., 2022*; *Gao et al., 2022*; *Patel and Jin, 2019*). We confirmed that *WT/SAVI* CD4 T cells were enriched with IFNγ$^+$ cells (*Figure 6A*). However, *WT/SAVI* mice have CD4 T cellpenia. Thus, the total numbers of spleen IFNγ$^+$ CD4 T cells were comparable in *WT/SAVI* and *AQ/SAVI* mice (*Figure 6A*). In contrast, the *HAQ/SAVI mice* had decreased IFNγ$^+$ CD4 T cells (*Figure 6A*).

The induction of Foxp3 expression in T-reg cells during ongoing autoimmune inflammation resolved inflammation and pathology in mice (*Hu et al., 2021*). CD4 T cellpenia depletes CD4 T-regs. Indeed, *WT/SAVI* mice had ~20-fold reduction of spleen FoxP3$^+$ T-regs compared to *AQ/SAVI* or *WT/WT* littermates (*Figure 6B*). The *HAQ/SAVI* mice also had ~10-fold more T-regs than the *WT/SAVI* littermate (*Figure 6B*).

## Discussion

This study, using the *HAQ, AQ, SAVI(N153S) Sting1* knock-in mice, reveals the in vivo significance and mechanism of STING1-mediated CD4 T cell death. *HAQ, AQ* alleles prevent CD4 T cellpenia, and increase/restore CD4 T-regs in SAVI mice. The results are consistent with previous finding that the impaired CD4 T cell proliferation by the *SAVI(V155M)* mutant could be rescued by the addition of the *HAQ* allele in vitro (*Cerboni et al., 2017*). STING1 has been increasingly implicated in inflammatory diseases such as nonalcoholic fatty liver disease, nonalcoholic steatohepatitis, cardiomyopathy, obesity, diabetes, neurodegenerative diseases, aging, and kidney injury, many of which are independent of type I IFNs (*Skopelja-Gardner et al., 2022*; *Bai and Liu, 2021*; *Gao et al., 2023a*). It is tempting to suggest that STING1 activation in CD4 T cells leads to CD4 T-regs depletion that break tissue tolerance and exacerbates tissue inflammation.

Human immunodeficiency virus (HIV) primarily infects CD4 T cells and might activate the STING1 pathway in CD4 T cells (*Monroe et al., 2014*; *Doitsh et al., 2014*; *Jakobsen et al., 2015*; *Silvin and Manel, 2015*; *Altfeld and Gale, 2015*; *Krapp et al., 2018*). The loss of CD4 T cells is the hallmark of untreated HIV infection *Gubser et al., 2019*; *Morou et al., 2019*, and the measurement of CD4 T cell count is a central part of HIV care. We found that *HAQ* and *AQ* CD4 T cells are resistant to STING1-mediated cell death. Mogensen and colleagues reported that *HAQ/HAQ* was enriched in HIV-infected long-term nonprogressors (*Nissen et al., 2018*). These *HAQ/HAQ* individuals had reduced inhibition of CD4 T cell proliferation and a reduced immune response to DNA and HIV (*Nissen et al., 2018*). It is likely that HIV infection activates STING1-cell death pathway in CD4 T cells. In *HAQ/SAVI* and *AQ/SAVI* mice, one copy of *HAQ, AQ* allele suppressed CD4 T cell death. *HAQ, AQ* carriers might have fewer HIV-induced CD4 T cell death, thus being long-term nonprogressors in HIV infection-induced Acquired immunodeficiency syndrome (AIDS; *Nissen et al., 2018*). Targeting STING1 to prevent CD4 T cell death might be a valid therapy for AIDS.

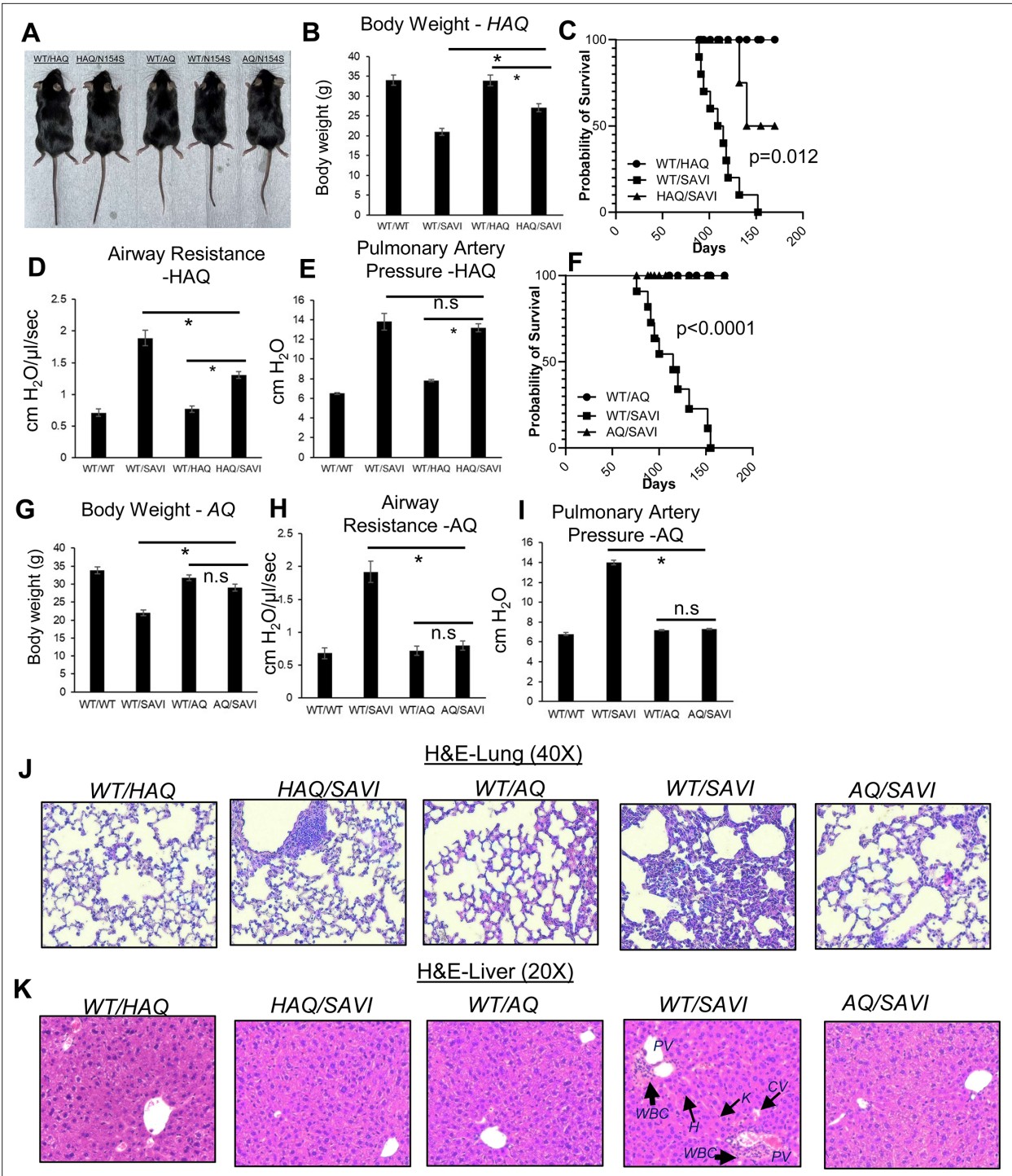

**Figure 4.** *HAQ* and *AQ* alleles prevent *SAVI(N153S)* disease in mice. (**A, B, G**) The size and body weight of *HAQ/SAVI, AQ/SAVI, WT/SAVI* and their littermates *WT/HAQ, WT/AQ* mice. (**D, E, H, I**). Airway resistance, and pulmonary artery pressure were determined as described in Materials and methods. (**C, F**). *HAQ/SAVI, AQ/SAVI, WT/SAVI* (10 mice/group), were monitored for survival by Kaplan-Meier. (**J, K**). Representative hematoxylin and eosin (H&E) staining of lung, liver sections from indicated mice. n=3–5 mice/group. Data are representative of three independent experiments. Graphs represent the mean with error bars indication s.e.m. p values are determined by one-way ANOVA Tukey's multiple comparison test. * p<0.05, **p<0.01. n.s.: not significant. (WBC): white blood cells; **H**: hepatocytes; K: Kupper cells; PV: portal vein; CV: central vein.

The online version of this article includes the following figure supplement(s) for figure 4:

**Figure supplement 1.** *HAQ, AQ* suppress lung myeloid cells infiltration in *SAVI(N153S)* mice.

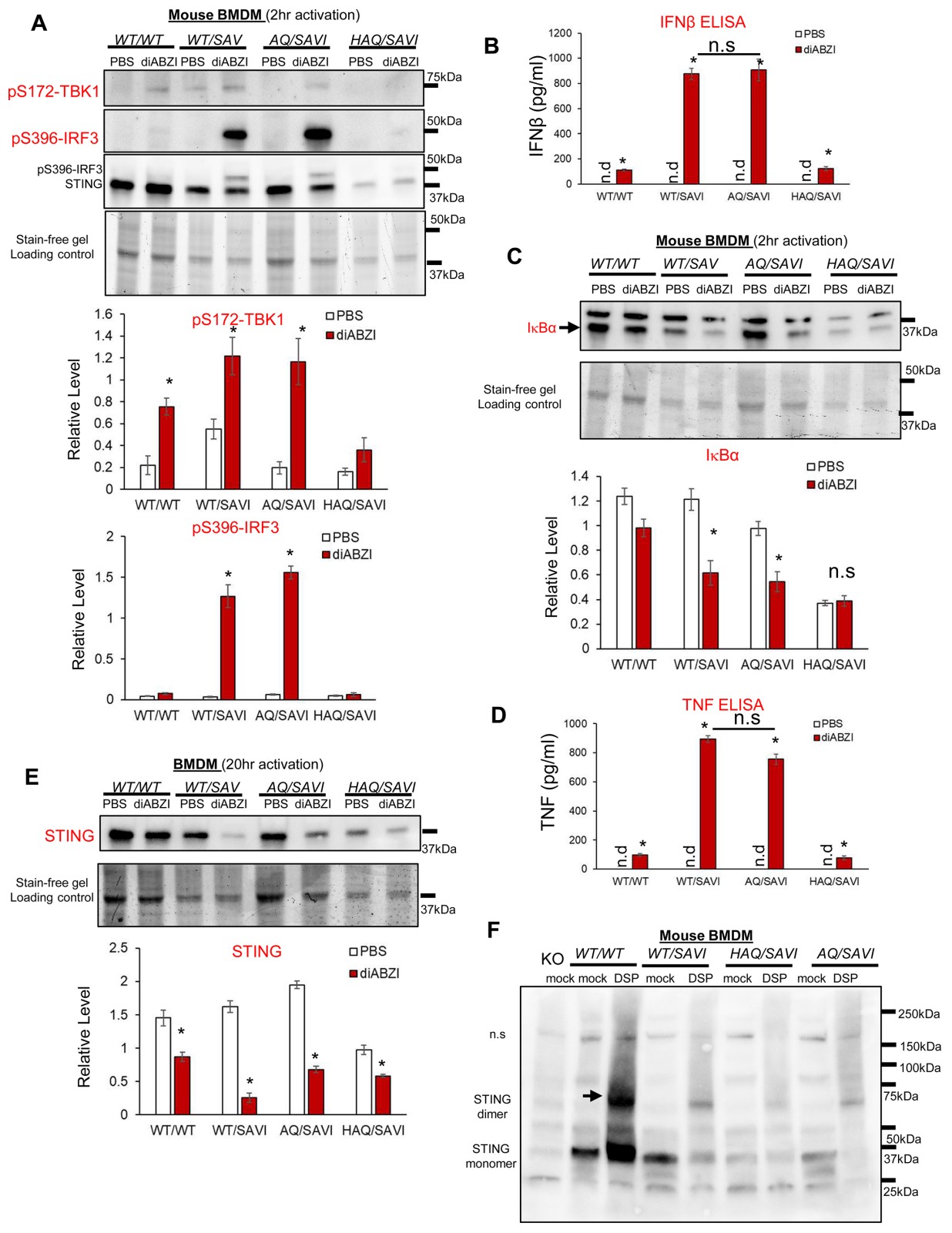

**Figure 5.** *AQ/SAVI(N153S)* cells had similar TBK1-IRF3, NF κ B activation and STING1 degradation as the *WT/SAVI(N153S)* cells. (**A, C**). BMDM from *WT/WT, WT/SAVI, HAQ/SAVI* and *AQ/SAVI* were treated with 100 ng/ml diABZi in culture for 2 hr. Cells were lysed and run on a 4~20% Mini-PROTEAN TGX Stain-Free Precast Gel. The blot was probed for phospho-Thr172-TBK1 antibody (CST, clone D52C2), phosphor-Ser 396-IRF3 (CST, clone 4D4G), STING1 (Proteintech, #19851–1-AP) and I κ Bα (CST, clone 44D4) antibody. (**E**). BMDM from *WT/WT, WT/SAVI, HAQ/SAVI* and *AQ/SAVI* were treated

*Figure 5 continued on next page*

*Figure 5 continued*

with 100 ng/ml diABZi in culture for 24 hr. Cells were lysed and run on a 4~20% Mini-PROTEAN TGX Stain-Free Precast Gel. The blot was probed for STING1 antibody (Proteintech, 19851–1-AP). (**B, D**). IFNβ and TNF were determined by ELISA in the cell supernatant from **E**. (**F**). BMDM from *WT/WT, WT/SAVI, HAQ/SAVI* and *AQ/SAVI* were treated with 400 μM cleavable chemical crosslinker DSP (Pierce, cat no: PG82081) in PBS for 1 hr at 4 °C. Cells were washed with PBS and lysed in RIPA buffer. Whole cell lysate was mixed with 4 x Laemmli Sample Buffer (BioRad, cat no 1610747) containing 5% 2-mercaptoethanol, heated at 95 °C for 10 min and, run on a 4~20% Mini-PROTEAN TGX Stain-Free Precast Gel. The blot was probed for STING1 antibody (Proteintech, 19851–1-AP). Densitometry was determined by ImageLab 5. Data are representative of three independent experiments. Graphs represent the mean with error bars indication s.e.m. p values are determined by unpaired student T-test (**A–E**) or one-way ANOVA Tukey's multiple comparison test (**D, B**). * p<0.05, **p<0.01, ***p<0.001. n.s.: not significant; n.d: not detected.

Activating the STING1 pathway is a promising strategy for cancer immunotherapy (*Hines et al., 2023*; *Samson and Ablasser, 2022*; *Liu et al., 2021*; *Zheng et al., 2020*; *Corrales et al., 2015*; *Fu et al., 2015*; *Barber, 2011*). Multiple STING1 agonists are in clinical trials (*Meric-Bernstam et al., 2023*; *Meric-Bernstam et al., 2022*). Recently, the safety issue emerged in some STING1 agonist trials (*Meric-Bernstam et al., 2023*; *Meric-Bernstam et al., 2022*). For example, in the STING1 agonist, ADU-S100, clinical trial, Grade 3/4 treatment-related adverse events were reported in 12.2% of 41 pretreated patients (NCT02675439) (*Meric-Bernstam et al., 2023*; *Meric-Bernstam et al., 2022*). The National Institutes of Health defines grade 3 as 'incapacitating; unable to perform usual activities; requires absenteeism or bed rest.' In a clinical trial using STING1 antibody-drug-conjugate (ADC) that conjugates diABZI to anti-HER2 Ab, a Grade 5 (fatal) serious adverse event was recorded and deemed related to the STING1-ADC (NCT05514717). SAVI disease, driven by overreacting STING1, is often fatal *Liu et al., 2014*. *AQ*, to a less degree, *HAQ*, suppress mortality in SAVI mice. Future STING1 clinical trials should be based on human *STING1* genotype to achieve safe and effective responses.

Mechanistically, apoptosis, pyroptosis, ferroptosis, necroptosis, and PANoptosis have all been reported in STING1-mediated cell death (*Kuhl et al., 2023*; *Jin et al., 2008*; *Gulen et al., 2017*; *Kabelitz et al., 2022*; *Murthy et al., 2020*; *Li et al., 2021*; *Song et al., 2022*; *Messaoud-Nacer et al., 2022*; *Tang et al., 2016*). Different cell types and STING1 agonists used likely contributed to the inconsistency and complexity. Here, we focused on lymphopenia in the SAVI mice that avoids ligand-dependent, non-physiological dosage in STING1-mediated cell death. *HAQ* and

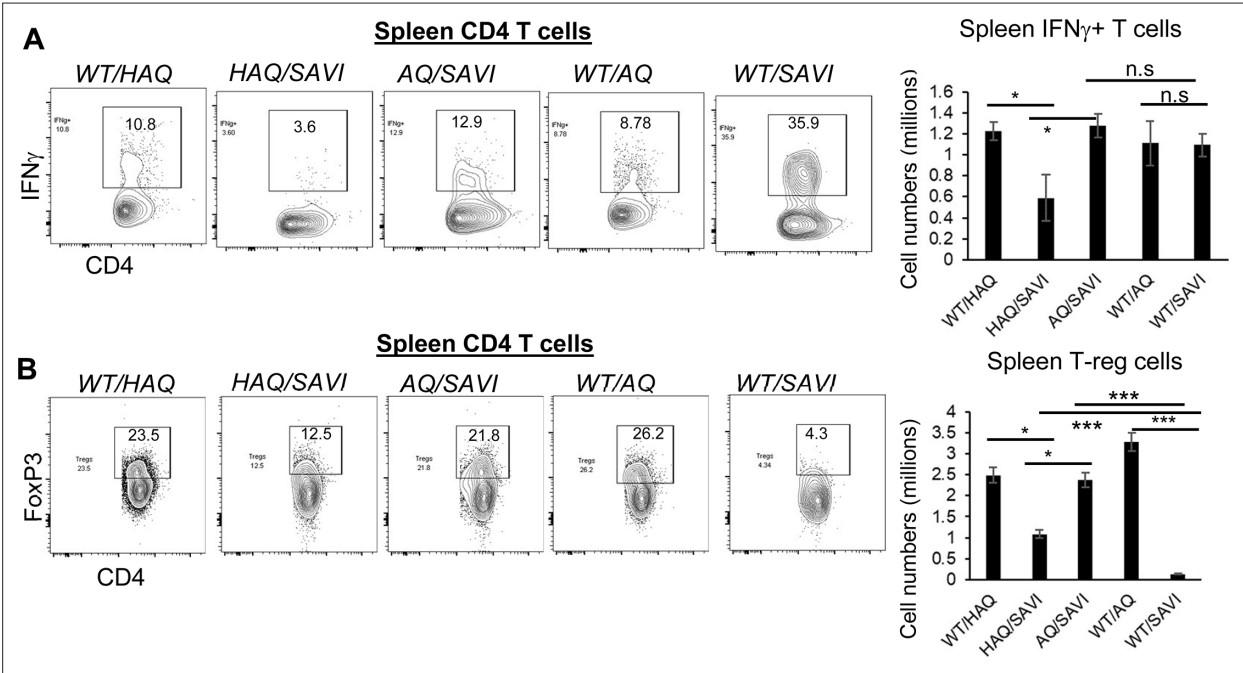

**Figure 6.** *HAQ/SAVI(N153S)* and *AQ/SAVI(N153S)* cells had 10-fold and 20-fold increased spleen T-regs compared to *WT/SAVI* mice. (**A**) Flow cytometry analysis of IFNγ producing spleen CD4+ T cells from *WT/HAQ, WT/AQ, WT/SAVI, HAQ/SAVI* and *AQ/SAVI* mice. (**B**) Flow cytometry analysis of CD4+ FoxP3+ spleen T-regs. n=3–5 mice/group. Data are representative of three independent experiments. Graphs represent the mean with error bars indication s.e.m. p values are determined by one-way ANOVA Tukey's multiple comparison test. * p<0.05, **p<0.01, ***p<0.001. n.s.: not significant.

*AQ* alleles could prevent CD4 T cellpenia in the SAVI mice strongly indicating that residue A230 or Q293 prevent STING1-mediated CD4 T cell death in vivo. Splenocyte from *Q293* mice were resistant to STING1 agonists-induced cell death ex vivo. Thus, it is likely that the Q293 residue is critical for STING1-mediate lymphopenia. Notably, Q293 is outside the C-terminal tail (CTT) (residues 341–379 of human STING1) critical for TBK1 recruitment and IRF3 phosphorylation (*Liu et al., 2015*) or miniCTT domain (aa343–354) (*Cerboni et al., 2017*), or the UPR motif (aa322–343) (*Wu et al., 2019*) important for T cells death in vitro. Further studies are needed to understand how the aa293 of STING1 mediates cell death in vivo. Noteworthy, *AQ/SAVI* cells had similar TBK1-IRF3, NFκB activation and STING1 degradation as the *WT/SAVI* cells. Yet, *AQ/SAVI* mice did not have CD4 T cellpenia as *WT/SAVI* mice suggeSting1 that the canonical STING1-TBK1-IRF3/NFκB pathway, likely STING1 oligomerization, is not sufficient for the induction of cell death at the physiological condition.

We used the *WT/N153S* knock-in SAVI mouse model that spontaneously develop lung inflammation, T cell cytopenia, and early mortality, mimicking pathological findings in human SAVI patients (*Warner et al., 2017*). Using the *WT/N153S* SAVI mouse model and human Jurkat T cell line, it was proposed that STING1 activation causes chronic ER stress and unfolded protein response, leading to T cell death by apoptosis (*Wu et al., 2019*). Furthermore, the study showed that crossing *WT/N153S* mice to the OT-I mice reduced ER stress and restored CD8[+], but not CD4[+], T cells (*Wu et al., 2019*). The restoration of CD8[+]T cells reduces inflammation and lung disease (*Wu et al., 2019*). However, human *WT/N154S* SAVI patients have normal CD8[+] T cells numbers (*Liu et al., 2014*), and primary human CD8[+] T cells are largely resistant to STING1-agonists-induced cell death ex vivo (*Figure 2A*; *Kuhl et al., 2023*). Thus, it is puzzling how restoring CD8[+] T cells can rescue SAVI phenotypes since the SAVI patients already have normal CD8[+] T cells numbers.

Finally, it is unexpected that both *HAQ* and *AQ* alleles are resistant to cell death. Our previous studies showed that the *HAQ* and *AQ* alleles have opposite functions (*Mansouri et al., 2022*). AQ-STING1, not HAQ-STING1, responds to CDNs (*Jin et al., 2011*; *Patel et al., 2017a*; *Mansouri et al., 2022*; *Sebastian et al., 2020*; *Yi et al., 2013*; *Patel et al., 2017b*; *Nissen et al., 2018*; *Ruiz-Moreno et al., 2018*; *Movert et al., 2023*). AQ mice are lean while *HAQ* mice are fat (*Mansouri et al., 2022*). Most importantly, *HAQ* was positively selected, while *AQ* was negatively selected, in modern humans outside Africans (*Mansouri et al., 2022*). Thus, the death pathway of STING1 is also distinct from the STING1 function that was naturally selected. It is worth noting that the *AQ* allele does better than the *HAQ* allele in suppressing SAVI disease. Thus, besides preventing cell death, additional mechanism by the *AQ* allele, for example fatty acid metabolism (*Mansouri et al., 2022*; *Vila et al., 2022*), is involved in curing SAVI.

## The limitations of the study

The poor transferability of mouse to humans is a major issue in STING1 research (*Meric-Bernstam et al., 2023*; *Meric-Bernstam et al., 2022*). The present study used *AQ/SAVI* and *HAQ/SAVI* mice. Confirmation is needed in humans with the identification and evaluation of people who are *AQ/SAVI*, *HAQ/SAVI*.

## Materials and methods
### Experimental design

The study was designed to reveal (i) the in vivo significance of the type I IFNs-independent, STING1-dependent cell death function; (ii) the interplay between common *STING1* alleles *HAQ, AQ* and the rare, gain-of-function SAVI STING1 mutation; (iii) the driver for the inflammatory SAVI disease. Mouse splenocytes, primary human lung cells, human THP-1 cells and *HAQ, AQ, SAVI* knock-in mice were used to establish the in vivo significance and human relevance. All the repeats were biological replications that involve the same experimental procedures on different mice. Where possible, treatments were assigned blindly to the experimenter by another individual in the lab. When comparing samples from different groups, samples from each group were analyzed in concert, thereby preventing any biases that might arise from analyzing individual treatments on different days. All experiments were repeated at least twice.

## Mice

*WT/SAVI(N153S)* mice were purchased from The Jackson Laboratory. *HAQ*, *AQ* mice were previously generated in the lab (*Patel et al., 2017a*; *Mansouri et al., 2022*). The *Q293* mice were generated by Cyagen Biosciences. Briefly, the linearized targeting vector was transfected into JM8A3.N1 C57BL/6 N embryonic stem cells. A positive embryonic stem clone was subjected to the generation of chimera mice by injection using C57BL/6 J blastocysts as the host. Successful germline transmission was confirmed by PCR sequencing. The heterozygous mice were bred to Actin-flpase mice [The Jackson Laboratory, B6.Cg-Tg (ACTFLPe)9205Dym/J] to remove the neo gene and make the *Q293* knock-in mouse. Age- and gender-matched mice (2–6 month old, both male and female) were used for indicated experiments. *WT/SAVI* (male) x *WT/HAQ* (female), *WT/SAVI* (male) x *WT/AQ* (female) breeders were set up to generate *HAQ/SAVI*, *AQ/SAVI* mice. Mice were housed at 22 °C under a 12-hr light-dark cycle with ad libitum access to water and a chow diet (3.1 kcal/g, Teklad 2018, Envigo, Sommerset, NJ) and bred under pathogen-free conditions in the Animal Research Facility at the University of Florida. Littermates of the same sex were randomly assigned to experimental groups. All mouse experiments were performed by the regulations and approval of the Institutional Animal Care and Use Committee at the University of Florida, IACUC202200000058.

## Reagent

Recombinant human IFNβ (R&D, cat no. 8499-IF-010/CF), diABZI (Invivogen, cat no. 2138299-34-8), 2'3'-cGAMP (Invivogen, cat. no. tlrl-nacga23-02), DMXAA (Invivogen, cat. no. tlrl-dmx), H151 (Invivogen, cat no. inh-h151), RpRpSS-Cyclic di-AMP (Biolog, cat no. C118), THP1-Dual KO-STING1 Cells (Invivogen, cat no. thpd-kostg). All other chemical inhibitors are from Selleckchem. Mouse TNF alpha ELISA Ready Set Go. (eBioscience, cat no. 88–7324). Mouse IFN-Beta ELISA Kit (PBI, cat no. 42400).

## Generation of THP-1 KO-STING1 cells stably expressing human *STING1* alleles

THP1-Dual KO-STING1 cells were purchased from Invivogen (thpd-kostg). These cells are guaranteed mycoplasma-free and authenticated by the vendor. THP1-Dual KO-STING1 Cells in six-well plate were transfected with 1 µg *STING1* plasmid (in pcDNA 3.1 vector) with Lipofectamine LTX and Plus Reagent (Invitrogen, cat no: A12621) according to the manufacturer's instructions. Transfecting Plasmid DNA into THP-1 Cells Using Lipofectamine LTX Reagent | Thermo Fisher Scientific - US. Forty-eight hours after the transfection, the cell medium was changed. G418 (1 mg/ml) was added to the culture to select STING1 expressing THP-1 cells. The G418-resistant cells were established and expanded.

## Histology

Lungs and livers were fixed in 10% formalin, paraffin-embedded, and cut into 4 µm sections. Lung, liver sections were then stained for hematoxylin-eosin. All staining procedures were performed by the histology core at the University of Florida. Briefly, tissue sectins were immersed Harris Hematoxylin for 10 s, then washed with tap water. Cleard sections were re-immersed in EOSIN stain for ~30 s. The sections were washed with tap water until clear, then dehydrate in ascending alcohol solutions (50%, 70%, 80%, 95% x 2, 100% x 2). Afterwards, the sections werer cleared with xylene (3–4 x). The sections were mounted on glass slide with permount organic mounting medium for visulization.

## Lung function

Pulmonary function was evaluated using an isolated, buffer-perfused mouse lung apparatus (Hugo Sachs Elektronik, March-Huggstetten, Germany), as previously described (*Cai et al., 2020*). Briefly, mice were anesthetized with ketamine and xylazine and a tracheostomy was performed, and animals were ventilated with room air at 100 breaths/min at a tidal volume of 7 µl/g body weight with a positive end-expiratory pressure of 2 cm $H_2O$ using a pressure-controlled ventilator (Hugo Sachs Elektronik, March-Huggstetten, Germany).

## Isolation of lung cells

Cells were isolated from the lung as previously described (*Mansouri et al., 2020*). The lungs were perfused with ice-cold PBS and removed. Lungs were digested in DMEM containing 200 µg/ml DNase

I (Roche, 10104159001), 25 µg/ml Liberase TM (Roche, 05401119001) at 37 °C for 2 hr. Red blood cells were then lysed and a single-cell suspension was prepared by filtering through a 70 µm cell strainer.

## BMDM activation

BMDMs were induced from mouse bone marrow cells cultured in RPMI 1640 (cat#11965; Invitrogen) with 10% FBS, 2 mM L-glutamine, 1 mM sodium pyruvate, 10 mM HEPES buffer, 1% nonessential amino acids, 50 mM 2-ME, 1% Pen/Strep, with 20 ng/ml M-CSF (Kingfisher, RP0407M) for 10 days (*Patel et al., 2017a*). STING1 agonists were added into the culture (no transfection or membrane permeabilization).

## Flow cytometry

Single-cell suspensions were stained with fluorescent-dye-conjugated antibodies in PBS containing 2% FBS and 1 mM EDTA. Surface stains were performed at 4 °C for 20 min. For intracellular cytokine or transcription factor staining of murine and human cells, cells were fixed and permeabilized with the Foxp3 staining buffer set (eBioscience, cat no 00-5523-00). Cells were washed and stained with surface markers. Cells were then fixed and permeabilized (eBioscience, cat no. 00-5523-00) for intracellular cytokine stain. Data were acquired on a BD LSRFortessa and analyzed using the FlowJo software package (FlowJo, LLC). Cell sorting was performed on the BD FACSAriaIII Flow Cytometer and Cell Sorter.

## Human lung explants

Human lung explants were procured at the Lung Transplant Center, Division of Pulmonary, Critical Care and Sleep Medicine, Department of Medicine, University of Florida. Donor and patients consent was obtained for a research protocol (UF IRB201902955-Treatment with IFNβ Induces Tolerogenic Lung Dendritic Cells in Human advanced lung disease). Healthy donor lungs were surgically removed postmortem, perfused, small pieces were cut from the right middle and lower lobes for research purpose, and stored in cold Perfadex at 4 °C for no more than 12 hr before processing. Ex planted lungs from emphysema lung transplant patients were stored in cold Perfadex at 4 °C for no more than 12 hr before the process. No lung explants were procured from prisoners.

## Statistical analysis

To gain statistical power, we employ three ~four mice/groups to characterize lung immunity. Ten mice/group to monitor animal health. The statistical justification for group size was calculated using the SAS program to calculate the animal numbers. The analysis was carried out using a standard error of 0.5 for immunological assays, and a power of 0.9. All data are expressed as means ± SEM. Statistical significance was evaluated using Prism 9.0 software. Comparisons between two groups were analyzed by performing an unpaired Student's $t$ test. Comparisons between more than two groups were analyzed by performing a one-way analysis of variance (ANOVA) with Tukey's multiple comparisons test.

## Materials availability statement

The Q293 mouse is available upon request via a Material Transfer Agreement.

## Acknowledgements

National Institutes of Health grant HL152163 (LJ).

## Additional information

### Funding

| Funder | Grant reference number | Author |
| --- | --- | --- |
| National Heart, Lung, and Blood Institute | HL152163 | Lei Jin |

| Funder | Grant reference number | Author |
|--------|------------------------|--------|

The funders had no role in study design, data collection and interpretation, or the decision to submit the work for publication.

## Author contributions

Alexandra a Aybar-Torres, Lennon A Saldarriaga, Ann T Pham, Data curation, Formal analysis, Investigation, Methodology; Amir M Emtiazjoo, Resources, Methodology; Ashish K Sharma, Data curation, Formal analysis, Investigation, Writing – review and editing; Andrew j Bryant, Data curation, Formal analysis, Supervision, Investigation, Writing – review and editing; Lei Jin, Conceptualization, Resources, Data curation, Formal analysis, Supervision, Funding acquisition, Validation, Investigation, Methodology, Writing - original draft, Project administration, Writing – review and editing

## Author ORCIDs

Lei Jin ⓘ https://orcid.org/0000-0001-6836-1142

## Ethics

The donor lung and data are from deceased individuals. The specimens or data are de-identified for the purpose of this study. Under these conditions, the human study is exempt from Federal regulation, under category 4 (E4) Human lung explants were procured at the Lung Transplant Center, Division of Pulmonary,Critical Care and Sleep Medicine, Department of Medicine, University of Florida. Donor and patients consent was obtained for a research protocol (UF IRB201902955-Treatment with IFNs; Induces Tolerogenic Lung Dendritic Cells in Human advanced lung disease).

All mouse experiments were performed by the regulations and approval of the Institutional Animal Care and Use Committee at the University of Florida, IACUC202200000058.

Reviewer #1 (Public Review): https://doi.org/10.7554/eLife.96790.3.sa1
Reviewer #2 (Public Review): https://doi.org/10.7554/eLife.96790.3.sa2
Author response https://doi.org/10.7554/eLife.96790.3.sa3

# Additional files

## Supplementary files

• MDAR checklist

## Data availability

Source data is deposited in Dryad https://doi.org/10.5061/dryad.m0cfxppcv.

The following dataset was generated:

| Author(s) | Year | Dataset title | Dataset URL | Database and Identifier |
|-----------|------|---------------|-------------|-------------------------|
| Jin L | 2024 | Data from: The common TMEM173 HAQ, AQ alleles rescue CD4 T cellpenia, restore T-regs, and prevent SAVI (N153S) inflammatory disease in mice | https://doi.org/10.5061/dryad.m0cfxppcv | Dryad Digital Repository, 10.5061/dryad.m0cfxppcv |

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
