## [Editor Report · eLife assessment]

This study describes **useful** mouse models of knock-ins of human STING1 variants and an assessment of these variants' action in mouse immune cells. While the implications of the variants in the inflammatory response are of significant interest, limitations are still found in the authors' interpretation and conclusions made, and the evidence for the conclusion remains **incomplete**.

---

## [Referee Report · Reviewer #1 (Public Review)]

Summary:

This manuscript by Aybar-Torres et al investigated the effect of common human STING1 variants on STING-mediated T cell phenotypes in mice. The authors previously made knock-in mice expressing human STING1 alleles HAQ or AQ, and here they established a new knock-in line Q293. The authors stimulated cells isolated from these mice with STING agonists and found that all three human mutant alleles resist cell death, leading to the conclusion that R293 residue is essential for STING-mediated cell death (there are several caveats with this conclusion, more below). The authors also bred HAQ and AQ alleles to the mouse Sting1-N153S SAVI mouse and observed varying levels of rescue of disease phenotypes with the AQ allele showing more complete rescue than the HAQ allele. The Q293 allele was not tested in the SAVI model. They conclude that the human common variants such as HAQ and AQ have a dominant negative effect over the gain-of-function SAVI mutants.

Strengths:

The authors and Dr. Jin's group previously made important observations of common human STING1 variants, and these knock-in mouse models are essential for understanding the physiological function of these alleles.

Weaknesses:

However, although some of the observations reported here are interesting, the data collectively does not support a unified model. The authors seem to be drawing two sets of conclusions from in vitro and in vivo experiments, and neither mechanism is clear. Several experiments need better controls, and these knock-in mice need more comprehensive functional characterization.

---

## [Referee Report · Reviewer #2 (Public Review)]

Aybar-Torres and colleagues utilize common human STING alleles to dissect the mechanism of SAVI inflammatory disease. The authors demonstrate that these common alleles alleviate SAVI pathology in mice, and perhaps more importantly use the differing functionality of these alleles to provide insight into requirements of SAVI disease induction. Their findings suggest that it is residue A230 and/or Q293 that are required for SAVI induction, while the ability to induce an interferon-dependent inflammatory response is not. This is nicely exemplified by the AQ/SAVI mice that have an intact inflammatory response to STING activation, yet minimal disease progression. As both mutants seem to be resistant STING-dependent cell death, this manuscript also alludes to the importance of STING-dependent cell death, rather than STING-dependent inflammation, in the progression of SAVI pathology. I believe this manuscript makes some important connections between STING pathology mouse models and human genetics that would contribute to the field.

---

## [Author Response]

The following is the authors’ response to the original reviews.

Summary Responses: Besides the *WT* allele, equivalent to the mouse *TMEM173* gene, the human *TMEM173* gene has two common alleles: the *HAQ* and *AQ* alleles carried by billions of people. The main conclusions and interpretation, summarized in the Title and Abstract, are (i) Different from the *WT TMEM173* allele, the *HAQ* or *AQ* alleles are resistant to STING activation-induced cell death; (ii) STING residue 293 is critical for cell death; (iii) *HAQ, AQ* alleles are dominant to the *SAVI* allele; (iv) One copy of the *AQ* allele rescues the SAVI disease in mice. We propose that STING research and STING-targeting immunotherapy should consider human *TMEM173* heterogeneity. These interpretations and conclusions were based on Data and Logic. We welcome alternative, logical interpretations and collaborations to advance the human *TMEM173* research.

**Public Reviews:**

**Reviewer #1 (Public Review):**
Summary:This manuscript by Aybar-Torres et al investigated the effect of common human STING1 variants on STING-mediated T cell phenotypes in mice. The authors previously made knock-in mice expressing human STING1 alleles HAQ or AQ, and here they established a new knock-in line Q293. The authors stimulated cells isolated from these mice with STING agonists and found that all three human mutant alleles resist cell death, leading to the conclusion that R293 residue is essential for STING-mediated cell death (there are several caveats with this conclusion, more below). The authors also bred HAQ and AQ alleles to the mouse Sting1-N153S SAVI mouse and observed varying levels of rescue of disease phenotypes with the AQ allele showing more complete rescue than the HAQ allele. The Q293 allele was not tested in the SAVI model. They conclude that the human common variants such as HAQ and AQ have a dominant negative effect over the gain-of-function SAVI mutants.Strengths:The authors and Dr. Jin's group previously made important observations of common human STING1 variants, and these knock-in mouse models are essential for understanding the physiological function of these alleles.Weaknesses:However, although some of the observations reported here are interesting, the data collectively does not support a unified model. The authors seem to be drawing two sets of conclusions from in vitro and in vivo experiments, and neither mechanism is clear. Several experiments need better controls, and these knock-in mice need more comprehensive functional characterization.(1) In Figure 1, the authors are trying to show that STING agonist-induced splenocytes cell death is blocked by HAQ, AQ and Q alleles. The conclusion at line 134 should be splenocytes, not lymphocytes. Most experiments in this figure were done with mixed population that may involve cell-to-cell communication. Although TBK1-dependence is likely, a single inhibitor treatment of a mixed population is not sufficient to reach this conclusion.

We greatly appreciate Reviewer 1's insights. We changed the “lymphocytes” to “splenocytes” (line 133) as suggested. We respectfully disagree with Reviewer 1’s comments on TBK1. First, we used two different TBK1 inhibitors: BX795 and GSK8612. Second, because BX795 also inhibits PDK1, we used a PDK1 inhibitor GSK2334470; Third, both BX795 and GSK8612 completely inhibited diABZI-induced splenocyte cell death (Figure 1B) (lines 128 – 133). The logical conclusion is “*TBK1 activation is required for STING-mediated mouse spleen cell death* ex vivo*”*. (line 117).

Our discovery that the common human *TMEM173* alleles are resistant to STING activation-induced cell death is a substantial finding. It further strengthens the argument that the *HAQ* and *AQ* alleles are functionally distinct from the *WT* allele 1-3. We wish to underscore the crucial message of this study-that *'STING research and STING-targeting immunotherapy should consider TMEM173 heterogeneity in humans*' (line 37), which has been largely overlooked in current STING clinical trials 4.

Regarding STING-Cell death, as we stated in the Introduction (lines 65-77). (i) STING-mediated cell death is cell type-dependent 5-7 and type I IFNs-independent 5,7,8. (ii) The in vivo biological significance of STING-mediated cell death is not clear 7,8. (iii) The mechanisms of STING-Cell death remain controversial. Multiple cell death pathways, i.e., apoptosis, necroptosis, pyroptosis, ferroptosis, and PANoptosis, are proposed 7,9,10. *SAVI/HAQ*, *SAVI/AQ* prevented lymphopenia and alleviated SAVI disease in mice. Thus, the manuscript provides some answers to the biological significance of STING-cell death in vivo, which is new. Regarding the molecular mechanism, splenocytes from *Q293/Q293* mice are resistant to STING cell death. The logical conclusion is that the amino acid 293 is critical for STING cell death (line 29).

Extensive studies are needed, beyond the scope of this manuscript, on how aa293 and TBK1 mediates STING-Cell death to resolve the controversies in the STING-cell death fields (*e.g.* apoptosis, necroptosis, pyroptosis, ferroptosis, and PANoptosis).

(2) Q293 knock-in mouse needs to be characterized and compared to HAQ and AQ. Is this mutant expressed in tissues? Does this mutant still produce IFN and other STING activities? Does the protein expression level altered on Western blot? Is the mutant protein trafficking affected? In the authors' previous publications and some of the Western blot here, expression levels of each of these human STING1 protein in mice are drastically different. HAQ and AQ also have different effects on metabolism (pmid: 36261171), which could complicate interoperation of the T cell phenotypes.

These are very important questions that require rigorous investigations that are beyond the scope of this manuscript. This manuscript, titled “*The common TMEM173 HAQ, AQ alleles rescue CD4 T cellpenia, restore T-regs, and prevent SAVI (N153S) inflammatory disease in mice*” does not focus on Q293 mice. We have been investigating the common human *TMEM173* alleles since 2011 from the discovery 11 , mouse model 1,3, human clinical trial 2, and human genetics studies 3. This manuscript is another step towards understanding these common human *TMEM173* alleles with the new discovery that *HAQ, AQ* alleles are resistant to STING cell death.

(3) HAQ/WT and AQ/WT splenocytes are protected from STING agonist-induced cell death equally well (Figure 1G). HAQ/SAVI shows less rescue compared to AQ/SAVI. These are interesting observations, but mechanism is unclear and not clearly discussed. E.g., how does AQ protect disease pathology better than HAQ (that contains AQ)? Does Q293 allele also fully rescue SAVI?

In this manuscript, Figure 6 shows *AQ/SAVI* had more T-regs than *HAQ/SAVI* (lines 251 – 261). In our previous publication on *HAQ, AQ* knockin mice, we showed that *AQ* T-regs have more IL-10 than *HAQ* T-regs 3. Thus, increased IL-10+ Tregs in *AQ* mice may contribute to an improved phenotype in *AQ/SAVI* compared to *HAQ/SAVI.* However, we are not excluding other contributions (e.g. metabolic difference) (lines 332-335). We are exploring these possibilities.

(4) Figure 2 feels out of place. First of all, why are the authors using human explant lung tissues? PBMCs should be a better source for lymphocytes. In untreated conditions, both CD4 and B cells show ~30% dying cells, but CD8 cells show 0% dying cells. This calls for technical concerns on the CD8 T cell property or gating strategy because in the mouse experiment (Figure 1A) all primary lymphocytes show ~30% cell death at steady-state. Second, Figure 2C, these type of partial effect needs multiple human donors to confirm. Three, the reconstitution of THP1 cells seems out of place. STING-mediated cell death mechanism in myeloid and lymphoid cells are likely different. If the authors want to demonstrate cell death in myeloid cells using THP1, then these reconstituted cell lines need to be better validated. Expression, IFN signaling, etc. The parental THP1 cells is HAQ/HAQ, how does that compare to the reconstitutions? There are published studies showing THP1-STING-KO cells reconstituted with human variants do not respond to STING agonists as expected. The authors need to be scientifically rigorous on validation and caution on their interpretations.

Figure 2 is necessary because it reveals the difference between mouse and human STING cell death, which is critical to understand STING in human health and diseases (lines 160-161). Figure 2A-2B showed that STING activation killed human CD4 T cells, but not human CD8 T cells or B cells. This observation is different from Figure 1A, where STING activation killed mouse CD4, CD8 T cells, and CD19 B cells, revealing the species-specific STING cell death responses. Regarding human CD8 T cells, as we stated in the Discussion (lines 323-325), human CD8 T cells (PBMC) are not as susceptible as the CD4 T cells to STING-induced cell death 8. We used lung lymphocytes that showed similar observations (Figure 2A). For Figure 2C, we used 2 WT/HAQ and 3 WT/WT individuals (lines 738-739). We generate HAQ, AQ THP-1 cells in STING-KO THP-1 cells (Invivogen,, cat no. thpd-kostg) (lines 380-387).

A recent study found that a new STING agonist SHR1032 induces cell death in STING-KO THP-1 cells expressing WT(R232) human STING 10 (line 182). SHR1032 suppressed THP1-STING-WT(R232) cell growth at GI50: 23 nM while in the parental THP1-STING-HAQ cells, the GI50 of SHR1032 was >103 nM 10. Cytarabine was used as an internal control where SHR1032 killed more robustly than cytarabine in the THP1-STING-WT(R232) cells but much less efficiently than cytarabine in the THP-1-STING-HAQ cells 10.

Our manuscript rigorously uses mouse splenocytes, human lung lymphocytes, THP-1 reconstituted with HAQ, AQ, and HAQ/SAVI, AQ/SAVI mice, to demonstrate that the common human HAQ, AQ alleles are resistant to STING cell death in vitro and in vivo.

We agree with Reviewer 1 that STING-mediated cell death mechanisms in myeloid and lymphoid cells may be different and likely contribute to the different mechanisms proposed in STING cell death research 7,9,10. Our study focuses on the in vivo STING-mediated T cellpenia.

(5) Figure 2G, H, I are confusing. AQ is more active in producing IFN signaling than HAQ and Q is the least active. How to explain this?

We stated in the Introduction that “*AQ* responds to CDNs and produce type I IFNs in vivo and in vitro 3,12,13 ”(line 92-93). We reported that the *AQ* knock in mice responded to STING activation 3. We previously showed that there was a negative natural selection on the *AQ* allele in individuals outside of Africa 3. 28% of Africans are *WT/AQ* but only 0.6% East Asians are *WT/AQ* 3. In contrast, the *HAQ* allele was positively selected in non-Africans 3. Investigation to understand the mechanisms and biological significance of these naturally selected human *TMEM173* alleles has been ongoing in the lab.

(6) The overall model is unclear. If HAQ, AQ and Q are loss-of-function alleles and Q is the key residue for STING-mediated cell death, then why AQ is the most active in producing IFN signaling and AQ/SAVI rescues disease most completely? If these human variants act as dominant negatives, which would be consistent with the WT/het data, then how do you explain AQ is more dominant negative than HAQ?

In this manuscript, Figure 6 shows *AQ/SAVI* had more T-regs than *HAQ/SAVI* (lines 251 – 261). In our previous publication on *HAQ, AQ* knockin mice, we showed that *AQ* T-regs have more IL-10 and mitochondria activity than *HAQ* T-regs 3. Nevertheless, we are not excluding other contributions (e.g. metabolic difference) by the *AQ* allele (lines 332-335). Last, we used modern human evolution to discover the dominance of these common human STING alleles. In modern humans outside Africans, *HAQ* was positively selected while *AQ* was negatively selected 3. However, *AQ* is likely dominant to *HAQ* because there is no *HAQ/AQ* individuals outside Africa. The genetic dominance of common human *TMEM173* allele is a new concept. More investigation is ongoing.

(7) As a general note, SAVI disease phenotypes involve multiple cell types. Lymphocyte cell death is only one of them. The authors' characterization of SAVI pathology is limited and did not analyze immunopathology of the lung.

Both radioresistant parenchymal and/or stromal cells and hematopoietic cells influence SAVI pathology in mice 14,15. Nevertheless, the lack of CD 4 T cells, including the anti-inflammatory T-regs, likely contributes to the inflammation in SAVI mice and patients 16. We characterized lung function, lung inflammation (Figure 4), lung neutrophils, and inflammatory monocyte infiltration (Figure S5) (lines 232-235).

(8) Line 281, the discussion on HIV T cell death mechanism is not relevant and over-stretching. This study did not evaluate viral infection in T cells at all. The original finding of HAQ/HAQ enrichment in HIV/AIDS was 2/11 in LTNP vs 0/11 in control, arguably not the strongest statistics.

Several publications have linked STING to HIV pathogenesis 17-22 (line 271). CD4 T cellpenia is a hallmark of AIDS. The manuscript studies STING activation-induced T cellpenia in vivo. It is not stretching to ask, for example, does preventing STING T cell death (e.g *HAQ, AQ* alleles) can restore CD4 T cell counts and improve care for AIDS patients?

**Reviewer #2 (Public Review):**
Aybar-Torres and colleagues utilize common human STING alleles to dissect the mechanism of SAVI inflammatory disease. The authors demonstrate that these common alleles alleviate SAVI pathology in mice, and perhaps more importantly use the differing functionality of these alleles to provide insight into requirements of SAVI disease induction. Their findings suggest that it is residue A230 and/or Q293 that are required for SAVI induction, while the ability to induce an interferon-dependent inflammatory response is not. This is nicely exemplified by the AQ/SAVI mice that have an intact inflammatory response to STING activation, yet minimal disease progression. As both mutants seem to be resistant STING-dependent cell death, this manuscript also alludes to the importance of STING-dependent cell death, rather than STING-dependent inflammation, in the progression of SAVI pathology. While I have some concerns, I believe this manuscript makes some important connections between STING pathology mouse models and human genetics that would contribute to the field.Some points to consider:(1) While the CD4+ T cell counts from HAQ/SAVI and AQ/SAVI mice suggest that these T cells are protected from STING-dependent cell death, an assay that explores this more directly would strengthen the manuscript. This is also supported by Fig 2C, but I believe a strength of this manuscript is the comparison between the two alleles. Therefore, if possible, I would recommend the isolation of T cells from these mice and direct stimulation with diABZI or other STING agonist with a cell death readout.

Please see the new Figure S3 for cell death by diABZI, DMXAA in Splenocytes from *WT/WT, WT/HAQ, HAQ/SAVI, AQ/SAVI* mice. The *HAQ/SAVI* and *AQ/SAVI* splenocytes showed similar partial resistance to STING activation-induced cell death (lines 214-216).

(2) Related to the above point - further exemplifying that the Q293 locus is essential to disease, even in human cells, would also strengthen the paper. It seems that CD4 T cell loss is a major component of human SAVI. While not co_mpletely necessary, repeating the THP1 cell death experiments from Fig 2 with a human T cell line would round out the study nicely._

We examined *HAQ, AQ* mouse splenocytes, *HAQ* human lung lymphocytes, THP-1 reconstituted with *HAQ, AQ,* and *HAQ/SAVI, AQ/SAVI* mice, to demonstrate that the common human *HAQ, AQ* alleles are resistant to STING cell death in vitro and in vivo. Additional human T cell line work does not add too much. We hope to conduct more human PBMC or lung lymphocytes STING cell death experiments from *HAQ, AQ* individuals as we continue the human *STING* alleles investigation.

(3) While I found the myeloid cell counts and BMDM data interesting, I think some more context is needed to fully loop this data into the story. Is myeloid cell expansion exemplified by SAVI patients? Do we know if myeloid cells are the major contributors to the inflammation these patients experience? Why should the SAVI community care about the Q293 locus in myeloid cells?

This is likely a misunderstanding. We use BMDM for the purpose of comparing STING signaling (TBK1, IRF3, NFkB, STING activation) by *WT/SAVI, HAQ/SAVI, AQ/SAVI*. Ideally, we would like to compare STING signaling in CD4 T cells from *WT/SAVI to HAQ/SAVI, AQ/SAVI* mice. However, *WT/SAVI* has no CD4 T cells. Doing so, we are making the assumption that the basic STING signaling (TBK1, IRF3, NFkB, STING activation) is conserved between T cells and macrophages.

(4) The functional assays in Figure 4 are exciting and really connect the alleles to disease progression. To strengthen the manuscript and connect all the data, I would recommend additional readouts from these mice that address the inflammatory phenotype shown in vitro in Figure 5. For example, measuring cytokines from these mice via ELISA or perhaps even Western blots looking for NFkB or STING activation would be supportive of the story. This would also allow for some tissue specificity. I believe looking for evidence of inflammation and STING activation in the lungs of these mice, for example, would further connect the data to human SAVI pathology.

Reviewer 2 suggests looking for evidence of inflammation and STING activation in the lungs of *HAQ/SAVI, AQ/SAVI*. We would like to elaborate further. First, anti-inflammatory treatments, e.g. steroids, DMARDs, IVIG, Etanercept (TNF), rituximab, Nifedipine, amlodipine, et al., all failed in SAVI patients 23. JAK inhibitors on SAVI had mixed outcomes (lines 55-58). Second, Figure S5 examined lung neutrophils and inflammatory monocyte infiltration. Interestingly, while *AQ/SAVI* mice had a better lung function than *HAQ/SAVI* mice (Figure 4D, 4E vs 4H, 4I), *HAQ/SAVI* and *AQ/SAVI* lungs had comparable neutrophils and inflammatory monocyte infiltration (Figure S5). Last, SAVI is classified as type I interferonopathy 23, but the lung diseases of SAVI are mainly independent of type I IFNs 24-27. The *AQ* allele suppresses SAVI in vivo. Understanding the mechanisms by which *AQ* rescues SAVI may lead to curative care for SAVI patients.

**Recommendations for the authors:**

**Reviewer #1 (Recommendations For The Authors):**
One suggestion is to streamline this study by focusing on STING-mediated cell death only in CD4 T cells. The authors can use in vitro PBMC isolated human T cells, ex vivo T cells from the knock-in mice, and in vivo T cells from the SAVI breeding. The current manuscript includes myeloid cell death, Tregs, complex SAVI disease pathology, which is too confusing and too complex to explain with the varying effect from the three human STING1 variants.

We sincerely appreciate Reviewer 1’s suggestion. The goal of our human *STING* alleles research has always been translational, i.e. improving human health. Even as a monogenetic disease, the SAVI pathology is still complex. For example, thought as a type I Interferonopathy, SAVI is largely independent of type I IFNs. Similarly, STING-activation-induced cell death, while contribute to SAVI, is not the whole story, as the Reviewer pointed out in the Comment 3 & 6 &7. *HAQ/SAVI* mice still died early and had lung dysfunction (Figure 4). In contrast, *AQ/SAVI* mice restore lifespan and lung function. We had Figure 6 show different T-regs between *AQ/SAVI* and *HAQ/SAVI* mice. In addition, *AQ* mice had more IL-10+ T-regs than *HAQ* mice 3. Therefore, we are excited about developing AQ-based curative therapy for SAVI patients (preventing cell death and inducing immune tolerance). Again, we thank the Reviewer for the suggestion. Additional research is ongoing.

**Reviewer #2 (Recommendations For The Authors):**
Minor points(1) Generation of THP1 cells with the human STING alleles is missing from methods.

We added the protocol in the methods (lines 380-387). THP-1 KO line stable expressing WT STING was first described by Weikang Tao’s group 10.

(2) Some abbreviations are not expanded (CDA).

CDA is expanded as cyclic di-AMP (e.g. line 375).

References.

(1) Patel, S. *et al.* The Common R71H-G230A-R293Q Human TMEM173 Is a Null Allele. *J Immunol* 198, 776-787 (2017).

(2) Sebastian, M. *et al.* Obesity and STING1 genotype associate with 23-valent pneumococcal vaccination efficacy. *JCI Insight* 5 (2020).

(3) Mansouri, S. *et al.* MPYS Modulates Fatty Acid Metabolism and Immune Tolerance at Homeostasis Independent of Type I IFNs. *J Immunol* 209, 2114-2132 (2022).

(4) Sivick, K. E. *et al.* Comment on "The Common R71H-G230A-R293Q Human TMEM173 Is a Null Allele". *J Immunol* 198, 4183-4185 (2017).

(5) Gulen, M. F. *et al.* Signalling strength determines proapoptotic functions of STING. *Nat Commun* 8, 427 (2017).

(6) Kabelitz, D. *et al.* Signal strength of STING activation determines cytokine plasticity and cell death in human monocytes. *Sci Rep* 12, 17827 (2022).

(7) Murthy, A. M. V., Robinson, N. & Kumar, S. Crosstalk between cGAS-STING signaling and cell death. *Cell Death Differ* 27, 2989-3003 (2020).

(8) Kuhl, N. *et al.* STING agonism turns human T cells into interferon-producing cells but impedes their functionality. *EMBO Rep* 24, e55536 (2023).

(9) Li, C., Liu, J., Hou, W., Kang, R. & Tang, D. STING1 Promotes Ferroptosis Through MFN1/2-Dependent Mitochondrial Fusion. *Front Cell Dev Biol* 9, 698679 (2021).

(10) Song, C. *et al.* SHR1032, a novel STING agonist, stimulates anti-tumor immunity and directly induces AML apoptosis. *Sci Rep* 12, 8579 (2022).

(11) Jin, L. *et al.* Identification and characterization of a loss-of-function human MPYS variant. *Genes Immun* 12, 263-269 (2011).

(12) Yi, G. *et al.* Single nucleotide polymorphisms of human STING can affect innate immune response to cyclic dinucleotides. *PLoS One* 8, e77846 (2013).

(13) Patel, S. *et al.* Response to Comment on "The Common R71H-G230A-R293Q Human TMEM173 Is a Null Allele". *J Immunol* 198, 4185-4188 (2017).

(14) Gao, K. M. *et al.* Endothelial cell expression of a STING gain-of-function mutation initiates pulmonary lymphocytic infiltration. *Cell Rep* 43, 114114 (2024).

(15) Gao, K. M., Motwani, M., Tedder, T., Marshak-Rothstein, A. & Fitzgerald, K. A. Radioresistant cells initiate lymphocyte-dependent lung inflammation and IFNgamma-dependent mortality in STING gain-of-function mice. *Proc Natl Acad Sci U S A* 119, e2202327119 (2022).

(16) Hu, W. *et al.* Regulatory T cells function in established systemic inflammation and reverse fatal autoimmunity. *Nat Immunol* 22, 1163-1174 (2021).

(17) Monroe, K. M. *et al.* IFI16 DNA sensor is required for death of lymphoid CD4 T cells abortively infected with HIV. *Science* 343, 428-432 (2014).

(18) Doitsh, G. *et al.* Cell death by pyroptosis drives CD4 T-cell depletion in HIV-1 infection. *Nature* 505, 509-514 (2014).

(19) Jakobsen, M. R., Olagnier, D. & Hiscott, J. Innate immune sensing of HIV-1 infection. *Curr Opin HIV AIDS* 10, 96-102 (2015).

(20) Silvin, A. & Manel, N. Innate immune sensing of HIV infection. *Curr Opin Immunol* 32, 54-60 (2015).

(21) Altfeld, M. & Gale, M., Jr. Innate immunity against HIV-1 infection. *Nat Immunol* 16, 554-562 (2015).

(22) Krapp, C., Jonsson, K. & Jakobsen, M. R. STING dependent sensing - Does HIV actually care? *Cytokine Growth Factor Rev* 40, 68-76 (2018).

(23) Liu, Y. *et al.* Activated STING in a vascular and pulmonary syndrome. *N Engl J Med* 371, 507-518 (2014).

(24) Luksch, H. *et al.* STING-associated lung disease in mice relies on T cells but not type I interferon. *J Allergy Clin Immunol* 144, 254-266 e258 (2019).

(25) Stinson, W. A. *et al.* The IFN-gamma receptor promotes immune dysregulation and disease in STING gain-of-function mice. *JCI Insight* 7 (2022).

(26) Warner, J. D. *et al.* STING-associated vasculopathy develops independently of IRF3 in mice. *J Exp Med* 214, 3279-3292 (2017).

(27) Fremond, M. L. *et al.* Overview of STING-Associated Vasculopathy with Onset in Infancy (SAVI) Among 21 Patients. *J Allergy Clin Immunol Pract* 9, 803-818 e811 (2021).